# Submicron-Size Emitters of the 1.2–1.55 μm Spectral Range Based on InP/InAsP/InP Nanostructures Integrated into Si Substrate

**DOI:** 10.3390/nano12234213

**Published:** 2022-11-27

**Authors:** Ivan Melnichenko, Eduard Moiseev, Natalia Kryzhanovskaya, Ivan Makhov, Alexey Nadtochiy, Nikolay Kalyuznyy, Valeriy Kondratev, Alexey Zhukov

**Affiliations:** 1International Laboratory of Quantum Optoelectronics, HSE University, 16 Soyuza Pechatnikov, St. Petersburg 190008, Russia; 2Ioffe Institute, Politehnicheskaya 26, St. Petersburg 194021, Russia; 3Center for Nanotechnologies, Alferov University, Khlopina 8/3, St. Petersburg 194021, Russia

**Keywords:** III–V nanostructures, InAsP/InP, silicon photonics, photoluminescence

## Abstract

We study photoluminescence of InP/InAsP/InP nanostructures monolithically integrated to a Si(100) substrate. The InP/InAsP/InP nanostructures were grown in pre-formed pits in the silicon substrate using an original approach based on selective area growth and driven by a molten alloy in metal–organic vapor epitaxy method. This approach provides the selective-area synthesis of the ordered emitters arrays on Si substrates. The obtained InP/InAsP/InP nanostructures have a submicron size. The individual InP/InAsP/InP nanostructures were investigated by photoluminescence spectroscopy at room temperature. The tuning of the emission line in the spectral range from 1200 nm to 1550 nm was obtained depending on the growth parameters. These results provide a path for the growth on Si(100) substrate of position-controlled heterojunctions based on InAs_1−x_P_x_ for nanoscale optical devices operating at the telecom band.

## 1. Introduction

Today, silicon is the most used semiconductor in the modern micro and nanoelectronic industry. However, despite the current level of and miniaturization of silicon technology, an efficient silicon laser, which is necessary for the needs of optoelectronics, is still far from being realized due to the indirect structure of the silicon band gap. In turn, many III–V compounds have a direct-gap structure and high efficiency of light emission. Therefore, in order to drive the further development of silicon photonics and to achieve its new functionality, a monolithic integration of III–V materials with well-developed silicon technology is necessary. 

The use of III–V nanostructures integrated with silicon is promising for creating devices for new integrated photonics [1,2,3,4], optical interconnects [5], and optoelectronic devices for high-speed signal processing [6,7]. At the same time, such integration is a difficult task due to the large lattice mismatch of most III–V materials with Si, a significant difference in thermal expansion coefficients, and a too high defect density for instrumental applications [8].

Currently, several competing approaches to the integration of III–V semiconductors with a silicon platform are being actively studied: the use of buffer layers [9,10,11], wafer bonding methods [12,13,14,15], selective growth methods [7,16,17,18,19,20], and also growth nanostructures using catalyst drops [21,22,23]. The most researched are approaches using wafer bonding methods. Their implementation implies the transfer of a finished III–V structure containing a highly efficient active region onto a silicon substrate with passive photonic structures. However, achieving high reliability, acceptable device yield, low cost, and the ability to integrate with complex optoelectronic integrated circuits remains a major challenge for this approach.

An alternative here is to grow III–V structures directly on silicon. As a rule, the effectiveness of such solutions is limited by the occurrence of defects in the active region of devices. Several recent works have investigated methods for filtering defects in such structures [8], the use of which, as a rule, negatively affects the compactness, complexity and cost of manufacturing such structures and the possibility of scaling to large areas.

Selective growth (SAG) and droplet growth methods are considered promising because they eliminate the use of expensive III–V substrates, complex wafer bonding technologies, and the difficulty of precise positioning when combining III–V and Si components at the micro level. In recent years, some groups have managed to obtain high-quality III-V structure nanostructures grown directly on Si, suitable for creating single laser structures [16,18,24,25]. In particular, SAG approaches have been used to create highly ordered arrays of nanostructures with precise positioning [8,18,26,27].

On the other hand, recently developed methods of epitaxial synthesis using droplets of group III metals make it possible to obtain III–V nanostructures on Si with a low defect density due to the small area of contact with the substrate and the relaxation of elastic stresses on the lateral surface. Significant progress has been made in the synthesis of ordered heterostructured III–V nanowires on a Si substrate [28,29]. More recently, a new epitaxial growth method using organometallic vapor phase epitaxy, called selective melt-based growth (MADSAG), has been introduced recently [30]. This method is a combination of growth elements, drop-induced group III element and SAG. The proposed approach combines the advantages of two methods: accurate positioning of nanostructures and growth selectivity, on the one hand, and a high degree of control over nucleation in the liquid phase, on the other hand. It was shown that the new approach makes it possible to obtain InP nanoinserts of high crystalline quality in a Si(100) substrate without using a buffer layer.

Further development of the proposed method was the formation of a semiconductor III-V heterostructure of controlled composition to obtain PL emission in the silicon transparency window (from 1.3 and 1.55 μm). In this work, we study the optical properties of such III-V heterostructures, formed by InP/InAsP/InP layers in the nanoinsertions synthesized by the MADSAG method in silicon. It is shown for the first time that it is possible to realize an ordered array of submicron-sized near-IR emitters with high structural and optical quality. By changing the growth parameters of the InAsP layer, one can control the emission wavelength of InP/InAsP/InP nanoinsertions in the range of 1.2–1.55 μm. 

## 2. Materials and Methods

III–V nanoinsertions were epitaxially grown inside the openings in Si(001) substrate. Technology aspects of the InP MADSAG growth were properly investigated in our previous work [30]. Here we used the growth in the optimized conditions for formation of InP/InAsP/InP heterostructure on Si(100). As in [30] prior to growth, Si(001) surface was covered by 100 nm-thick SiN_x_ mask. The arrays of 200 nm wide holes in the mask were defined using the deep ultraviolet lithography. The distance between the centers of the holes was 600 or 800 nm. Next, deep-reactive ion etching with SF_6_ chemistry was employed to etch isotopically 300 nm deep holes inside Si. The sidewalls of the holes were covered with SiN_x_ and then, at the bottom of the holes the {111} planes of Si were opened using KOH wet etching (Figure 1a). The dry etch leaves the rough and defective surface at the bottom of the hole, which can cause polycentric nucleation and formation of various types of defects in the III–V material. Additional KOH wet etch leaves flat {111} planes behind which is necessary to have more controllable nucleation and high crystalline quality [31]. The epitaxial growth started with the formation of InP layer inside the holes in Si (001) using the MADSAG approach. For the InP layer deposition trimethylindium (TMIn) and phosphine (PH_3_) were used. The MADSAG approach implies formation of the In-rich melt at the bottom of the hole which is then crystallized to InP during the annealing under 90 sccm flux of PH_3_ at substrate temperature of 600 °C, Figure 1b. Next, the InAs_x_P_1−x_ layer was formed, for which the supply of PH_3_ was stopped, and the arsine (AsH_3_) flux was supplied. The interaction of arsenic with the InP surface at selected temperatures (500 °C) resulted in the replacement of phosphorus by arsenic in the upper layers and the formation of an InAs_x_P_1−x_ layer (Figure 1c). For the InAs_x_P_1−x_ layer formation substrate temperature was set 500 °C. At the final stage (Figure 1d), an InP layer was formed covering the structure, which was formed similarly to the initial InP. The formed nanostructures were studied via scanning electron microscope Zeiss Supra 25 at 28 kV (SEM, Carl Zeiss AG, Oberkochen, Germany, 28kV). SEM contrast in secondary electrons was obtained at an accelerating voltage of approximately 30 kV. SEM image of a cross section of an InP/InAs_x_P_1−x_/InP heterostructure in silicon is presented in (Figure 1e).

Samples S1 and S2 were formed with a single layer of InAs_x_P_1−x_ as described above but in Samples S1 for InAs_x_P_1−x_ layer formation we performed the annealing under 175 sccm of AsH_3_ flux during 180 s and in Samples the annealing time was 60 s to tune emission wavelength. In Sample S3 we used growth parameters of the sample S2, but 5 InAsP layers were formed separated by an InP layer. The growth parameters are presented in Table 1.

PL maps and spectra were measured using an Integra Spectra (NT-MDT, Zelenograd, Russia) confocal microscope at room temperature. The Nd:YLF laser operating in continuous mode (527 nm wavelength) was used for excitation. The excitation laser beam was focused using a 100× objective (M Plan APO NIR, Mitutoyo, Japan) with a numerical aperture NA = 0.5. The pump power density can be varied from 0.2 to 230 kW/cm^2^. The same objective was used to collect the photoluminescence signal of InP/InAs_x_P_1−x_/InP nanostructures. The scanning over the surface was performed with a set of mirrors. The emission was directed to the entrance slits of a monochromator (MS5204i, Sol Instruments, Republic of Belarus) using mirrors. Detection was performed using a cooled InGaAs CCD array (iDus, Andor, UK). 

## 3. Results and Discussion

Figure 2a,b shows plan-view and cross-section, respectively, SEM images obtained from sample S1 with a distance between the centers of the etch pits of 800 nm. It can be seen that the growth of the InP/InAs_x_P_1−x_/InP material in the pits proceeds unevenly. Most of the pits in this sample turned out to be unfilled, and only in some pits a material fills in the pit in silicon. A decrease in the distance between the pits to 600 nm led to the even smaller filling of the pits, and therefore we did not study that sample further.

A confocal microscope was used to study the photoluminescence (PL) of the resulting nanostructures. At a distance between the centers of adjacent pits of 800 nm, it is possible to locally study the PL of free-standing nanoinsertion. The PL spectra were mapped from the area of the sample marked with a square in Figure 2a. In the PL intensity distribution measured in the spectral range of 800–1650 nm (Figure 2c), there are bright spots, which correspond to the positions of the large-volume nanoinsertion visible in the SEM image and indicated as points 1, 2 and 3 in Figure 2a. Point 4 corresponds to a nanoinsertion of a small volume, point 5 is located on the surface of the sample between the pits.

The PL spectra obtained at points 1–5 are shown in Figure 3a. In the spectra obtained at points 1–4, an intense line is observed, with a spectral position of the maximum at ~915 nm, which corresponds to the emission of InP in the sphalerite phase [32]. On the spectra obtained at points 1–3 there is an additional broad line located from 1300 to 1650 nm with a maximum near 1.45 μm. We associate this line with the emission of the narrow-bandgap InAs_x_P_1−x_ insertion in the InP/InAs_x_P_1−x_/InP heterostructure. The absence of this line at point 4 indicates that the formation of a complete InP/InAs_x_P_1−x_/InP heterostructure occurs only in some pits, which are visualized in SEM as large-volume nanoinsertions. At point 5, the spectrum demonstrates an almost zero intensity over the entire wavelength range, which indicates the absence of III–V material deposition on the Si_3_N_4_ surface, and all material from this region is collected in pits. The inhomogeneity of the collection of this material into the pits leads to the inhomogeneity of the InP/InAs_x_P_1−x_/InP nanostructures themselves, which explains the difference in the position of the maximum of the PL line at points 1–3 in Figure 2a.

The spectral position of the maximum PL intensity in the spectra obtained at points 1–3 varies from 1430 nm to 1460 nm (photon energy ~0.849—0.867 eV). One can estimate the average concentration of arsenic (x) in the InAs_x_P_1−x_ alloy at these points using the expression, which neglects the quantum-size effect:(1)$$E_{G}\left(x\right)=x\;E_{\mathrm{InAs}}+\left(1-x\right)\;E_{\mathrm{InP}}-C_{\mathrm{InAsP}}\;x\;\left(1-x\right)$$
where $E_{\mathrm{InAs}}$ and $E_{\mathrm{InP}}$ of 0.354 and 1.344 eV are the band gaps of InAs and InP, respectively, $C_{\mathrm{InAsP}}$ = 0.1 eV is the bowing parameter [33]. The average concentration of arsenic in the sample S1 is thus estimated to be ~45–47%. The observed variation in the position of the PL intensity maximum within 30 nm from one nanoinsertion to another (1430 nm to 1460 nm) can be a consequence of fluctuations in the average mole fraction of InAs just within only 3%, due to a rather strong dependence of the InAs_x_P_1−x_ bandgap on the x composition. The PL spectra from one InP/InAs_x_P_1−x_/InP nanoinsertion of sample S1 obtained at different optical pump powers at room temperature are shown in Figure 3b. It can be seen that as the optical pump power increases, the PL maximum shifts to shorter wavelengths, which can be explained by the inhomogeneity of the composition of the InAs_x_P_1–x_ nanoinsertion and its limited volume. At a low pump power, the lower energy states in the InAs_x_P_1−x_ are predominantly occupied, with an increase in the carrier concentration, the higher-energy states are gradually filled and the PL maximum is shifted to the short-wavelength region. In addition, due to the limited volume of the InAs_x_P_1−x_ insert, with an increase in the optical pump power, the increase in the intensity of the InAs_x_P_1−x_ line saturates, which is not observed for the InP spectral line.

A SEM image of the surface of sample S2, in which the annealing time during the supply of AsH_3_ was decreased compared to sample S1, is shown in Figure 4a. The SEM images show a periodic structure (the distance between the centers of the pits was 800 nm) as in the S1 structure. It is also seen that, again, in most of the pits, the InP/InAs_x_P_1−x_/InP material is located inside the pit, and only at a few points the formation of InP/InAs_x_P_1−x_/InP is visible. PL intensity distribution over the same area of the sample is shown in Figure 4b for the spectral range of InAsP emission (1150–1400 nm). One can observe areas of bright PL intensity, which position corresponds on the SEM image of sample S2 to the large-volume nanoinsertion indicated by points 1, 2 and 3.

The emission spectra obtained in the whole 800–1650 nm interval at points of bright intensity are shown in Figure 5. The spectra show a line corresponding to carrier recombination in the InP and a broad line with a maximum at approximately 1200 nm, which we associate with formed InAs_x_P_1−x_ material. The large spectral width of the line indicates fluctuations of InAs_x_P_1−x_ composition. The shift of the InAs_x_P_1−x_ spectral line to the short-wavelength region compared to sample S1 is due to the lower flux of the As precursor and thus the lower InAs mole fraction. The exposure time in arsine vapor as well as its flow highly affect the substitution of phosphorus by arsenic [34] in InP. For example, in [35] the formation of In(P)As quantum dots upon exposure to an As flow of the InP(311)B surface was studied. The PL wavelength of the formed quantum dots strongly depended on the substrate temperature and exposure duration and varied from ~1200 to ~1600 nm, which well coincides with our results.

The position of the PL maximum in the studied nanostructure varies in the spectral range 1193–1220 nm (1.039 eV–1.016 eV). From these data, one can similarly, as was done for sample S1, determine the spread of the average composition for arsenic in the formed nanoinsertions, which corresponds to an arsenic concentration from 29% to 31%. The resulting spread of arsenic concentration in the structure S2 does not exceed 3%, which indicates a high homogeneity and reproducibility of the technological process. 

Despite the fact that the initial pits in Si are identical to each other, the formed nanoinsertions of samples S1 and S2 show fluctuations in shapes and sizes. The observed non-uniformity in the distribution of the InP/InAs_x_P_1−x_/InP material over different pits we associate with the effect of the Ostwald ripening process (the formation of larger nanoinsertions from the material of the smaller ones) [30]. To reduce this effect in sample S3, the amount of material was increased by 5-fold repetition of the steps of forming the InAs_x_P_1−x_/InP layers. Figure 6a shows a SEM image with a top-view of this sample. It can be seen that in this structure, all the nanoinsertions are formed and there are no unfilled pits. Figure 6b shows a PL map in the spectral range from 1150 to 1400 nm in the same area of the sample. 

The periodic structure of the emitters is visible now, the position of which corresponds to the position of the InAs_x_P_1−x_/InP nanostructures in silicon. Spectra from InAs_x_P_1−x_/InP nanostructures marked with points 1, 2 and 3 are presented in Figure 7a. The spectra obtained from single InAs_x_P_1−x_/InP emitters are dominated by a bright spectral line, with a maximum near 1200 nm. The spectrum also contains a less intense line of InP luminescence. The PL intensity of the InAs_x_P_1−x_ line varies from point to point, which is most likely due to the uneven redistribution of the material between the etch pits. With an increase in the amount of deposited material in sample S3 compared to the previous samples, the intensity of the InAs_x_P_1−x_ PL line increases. With an increase in the optical pump power (Figure 7b), the intensity of this line does not saturate up to at least 230 kW/cm^2^ (inset to Figure 7b). This is in contrast with sample S1, where a bend is observed. The scatter in As composition (x) in sample S3, which we also estimated using formula (1), also does not exceed 3%.

## 4. Conclusions

In this work, we study InP/InAs_x_P_1−x_/InP nanostructures with submicron lateral size formed by the molten alloy driven selective-area growth (MADSAG) method in silicon. This novel method makes it possible to grow In(As)P nanoinsertions in silicon with a minimum number of defects due to close to equilibrium thermodynamic conditions of crystallization in a single drop. Using the method, the III–V semiconductor heterostructures of controlled composition were formed in Si with room-temperature emission in the Si transparency window from 1.2 to 1.55 μm. The optical properties of the obtained structures were studied, depending on the deposition parameters. It is shown that the proposed method makes it possible to form an ordered array of submicron near-IR emitters with high structural and optical quality. The variation of arsenic concentration in the InAs_x_P_1−x_ alloy from point to point within the array is as low as 3%. The observed variation of the PL maximum position in the studied nanostructure is due to the strong band gap composition dependence. By tiny changes of the MADSAG deposition conditions, one can control the emission wavelength of InP/InAs_x_P_1−x_/InP nanoinsertions from 1200 to 1550 nm. To obtain equal redistribution of III–V materials in pits and therefore more homogeneous PL intensity from pit to pit, additional research is required. The epitaxial growth is strongly affected by the diffusion of adatoms over the SiN_x_ mask surface. In its turn, the diffusion length depends on the substrate temperature and V-to-III ratio. Thus, studies of the growth parameters (temperature, V-to-III ratio, growth rate and growth interruptions) on uniformity are planned.

## Figures and Tables

**Figure 1 nanomaterials-12-04213-f001:**
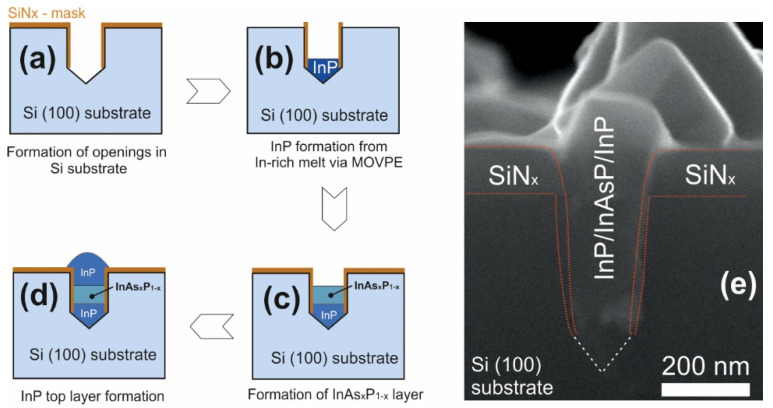
(**a**–**d**)—scheme of the main stages of the formation of InP/InAs_x_P_1−x_/InP nanoinclusions in silicon using the MADSAG method, (**e**)—SEM image of a cross section of an InP/InAs_x_P_1−x_/InP heterostructure in silicon.

**Figure 2 nanomaterials-12-04213-f002:**
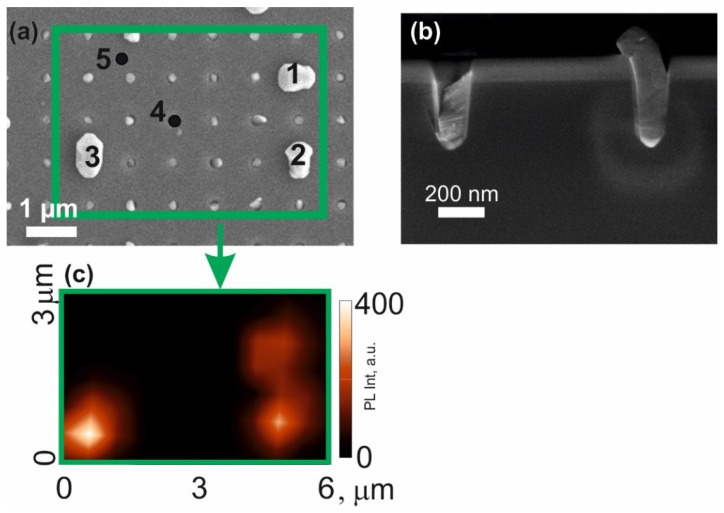
(**a**)—SEM obtained from the top of sample S1; (**b**)—SEM image of the cross-section of the sample S1; (**c**)—map of the distribution of photoluminescence intensity in the spectral range from 800 to 1650 nm in the area of the sample marked with a rectangle in (**a**).

**Figure 3 nanomaterials-12-04213-f003:**
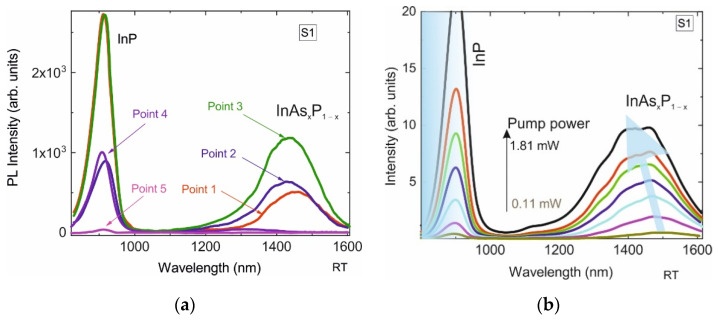
(**a**)—PL spectra of sample S1 obtained at points 1–5 at a pump power density of 127 kW/cm^2^. (**b**)—PL spectra from single InP/InAs_x_P_1−x_/InP nanoinsertion in sample S1 obtained at different optical pump powers (0.11 mW (14 kW/cm^2^)–1.81 mW (230 kW/cm^2^)).

**Figure 4 nanomaterials-12-04213-f004:**
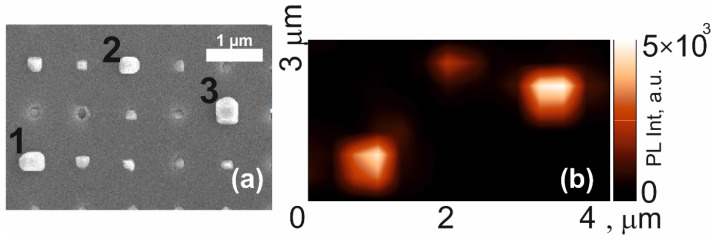
(**a**)—SEM image obtained from the sample S2, (**b**)—map of the PL intensity in the spectral range from 1150 to 1400 nm of the same section of the sample.

**Figure 5 nanomaterials-12-04213-f005:**
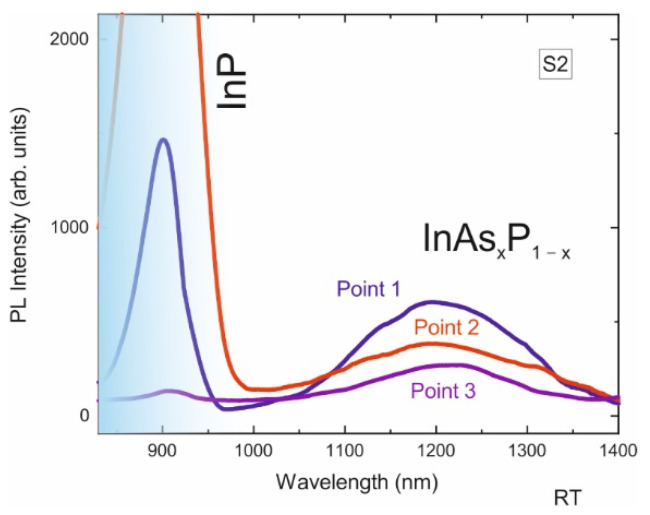
PL spectra obtained at points 1–3 of sample S2 at a pump power density of 127 kW/cm^2^.

**Figure 6 nanomaterials-12-04213-f006:**
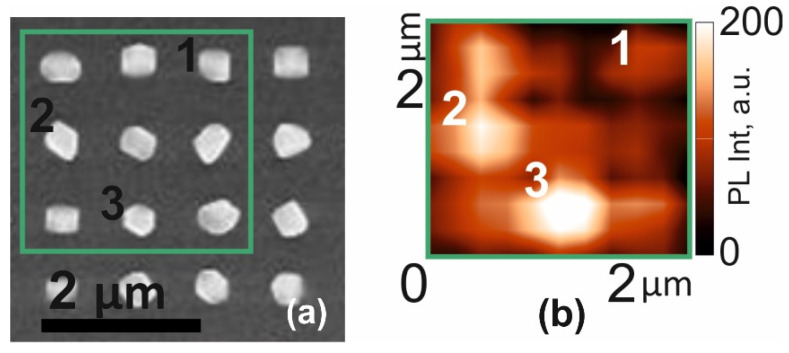
(**a**)—SEM image obtained from above sample S3; (**b**)—map of the photoluminescence intensity distribution in the spectral range from 1150 to 1400 nm in the sample area highlighted by the rectangle in (**a**).

**Figure 7 nanomaterials-12-04213-f007:**
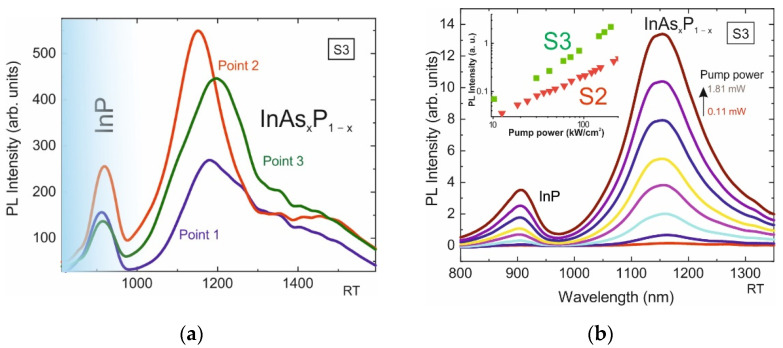
(**a**)—PL spectra of sample S3 obtained at points 1–3 at a pump power density of 127 kW/cm^2^. (**b**)—PL spectra from single InP/InAs_x_P_1−x_/InP nanoinsertion in sample S3 obtained at different optical pump powers (0.11 mW(14kW/cm^2^)–1.81mW(230kW/cm^2^); inset: Pl intensity of the samples S2 and S3 versus optical pump power.

**Table 1 nanomaterials-12-04213-t001:** Description of the Samples S1–S3.

Sample	Annealing Time under 175 sccm of AsH_3_ Flux	Number of InAs_x_P_1−x_ Layers
S1	180 s	1
S2	60 s	1
S3	60 s	5

## Data Availability

The data presented in this study are available on request from the corresponding author. The data are not publicly available due to the author’s readiness to provide it on request.

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
