# Peer review of "Submicron-Size Emitters of the 1.2–1.55 μm Spectral Range Based on InP/InAsP/InP Nanostructures Integrated into Si Substrate"

_nanomaterials, 2022, doi:10.3390/nano12234213_

Round 1

Reviewer 1 Report

This manuscript reports the photoluminescence properties of the InP/InAsP/InP nanostructures synthesized via MADSAG method. The manuscript is unable to emphasize the research gap and seems to have an extension work of the previously published work (Nanoscale, 2020, 12, 23780). Moreover, the experimental information provided in the manuscript is not sufficient and clear to justify the novel outcome of the research work. Reviewer does not recommend publishing this manuscript in the nanomaterials in its current form. Additional comments are also included.

1.       Abstract is not very clear and does not provide sufficient information on the research outcomes and application of the work.

2.       Authours need to improve the introduction section as it does not emphasize on the novelty of the work. Also, authours need to cite more recent work in this field, I available.

3.       The authors tried to synthesize the S(111) orientation inside the Si(001). However, authors have not provided any discussion as why it was necessary to make these structures.

4.       Figure 1 is not clear and concise to explain the synthesis process. Also, the experimental section explaining the synthesis of InP/InAsP/InP nanostructures is very confusing and open ended.

5.       The Authors need to clearly define the S1-S3. Providing a Table having all the sample details tabulated is suggested.

6.       Scale bars are not provided in the Figure 2(a).

7.       The figures are not very well connected in the discussion section.

Author Response

We are thankful to the Reviewer for this comment. Please, find a point-by-point response in the attached file.

Reviewer 2 Report

This paper reports a study on InP/InAsP/InP nanostructures with submicron lateral size using the molten alloy-driven selective-area growth (MADSAG) method on silicon, and it concludes an ordered array of submicron near-IR emitters forms with high structural and optical quality. The text is well written in general, except rephrasing a few phrases would be needed. I would recommend this manuscript be considered for publishing in the nanomaterials after a major correction.

The following should be answered or changed in the manuscript:

Monolithic integration of III-V on silicon is largely demanding for the modern electronic and photonic industry.  The authors also emphasize this in the introduction, “integration with silicon of such III-V materials as InP or GaAs could result in photonic integrated circuits (PICs) with optically active elements (amplifiers, lasers, etc.) with reduced footprint, lower power consumption and reduced total cost of its production”. The photonic active elements namely amplifiers and lasers need about tens to hundreds of microns of high-quality III-V on silicon.  Authors need to describe how this work could help towards this target. Addressing the technological issues, and how this work can be extended toward the monolithic integration of lasers and amplifiers on silicon requires a much larger area of growth.

I regret that there are no technical suggestions to obtain a more homogeneous emitter reported in the paper, and controlling the distribution of material between the pits is set for the near future in the conclusion. The authors should include a discussion section to describe the strategy to improve the methods.

Author Response

Also we highly appreciate Referee’s time and careful reading of the manuscript. Please find our point-by-point response in the attached file.

Round 2

Reviewer 2 Report

The manuscript has been improved and some extra explanations have been provided in the manuscript. The manuscript can be published in Nanomaterials considering a minor correction.

-The unit of optical power density should be modified in figures refer to PL measurements.
